# Frequency-related viscoelastic properties of the human incisor periodontal ligament under dynamic compressive loading

Bin Wu[1]☯, Panjun Pu[2]☯, Siyu Zhao[2], Iman Izadikhah[2], Haotian Shi[2], Mao Liu[2], Ruxin Lu[3], Bin Yan(ID)[2]*, Songyun Ma[4], Bernd Markert[4]

**1** College of Mechanical and Electronic Engineering, Nanjing Forestry University, Nanjing, China, **2** Jiangsu Key Laboratory of Oral Diseases, Department of Orthodontics, Affiliated Hospital of Stomatology, Nanjing Medical University, Nanjing, China, **3** College of Mechanical Engineering, Southeast University, Nanjing, China, **4** Institute of General Mechanics, RWTH-Aachen University, Aachen, Nordrhein-Westfalen, Germany

☯ These authors contributed equally to this work.
* byan@njmu.edu.cn

**Data Availability Statement:** All relevant data are within the paper and its Supporting Information files.

## Abstract

Studies concerning the mechanical properties of the human periodontal ligament under dynamic compression are rare. This study aimed to determine the viscoelastic properties of the human periodontal ligament under dynamic compressive loading. Ten human incisor specimens containing 5 maxillary central incisors and 5 maxillary lateral incisors were used in a dynamic mechanical analysis. Frequency sweep tests were performed under the selected frequencies between 0.05 Hz and 5 Hz with a compression amplitude that was 2% of the PDL's initial width. The compressive strain varied over a range of 4%-8% of the PDL's initial width. The storage modulus, ranging from 28.61 MPa to 250.21 MPa, increased with the increase in frequency. The loss modulus (from 6.00 MPa to 49.28 MPa) also increased with frequency from 0.05 Hz– 0.5 Hz but remained constant when the frequency was higher than 0.5 Hz. The tan$\delta$ showed a negative logarithmic correlation with frequency. The dynamic moduli and the loss tangent of the central incisor were higher than those of the lateral incisor. This study concluded that the human PDL exhibits viscoelastic behavior under compressive loadings within the range of the used frequency, 0.05 Hz– 5 Hz. The tooth position and testing frequency may have effects on the viscoelastic properties of PDL.

## Introduction

The periodontal ligament (PDL) serves as a connective tissue, linking and securing the tooth root to the alveolar bone. It is capable of transmission and resorption of forces induced by mastication and other tooth contacts [1]. It also plays an important role in the process of bone remodeling, and understanding its structural and biomechanical properties is a leading factor in perceiving tooth movement. The PDL consists of collagen fibers and a liquefied ground substance of proteoglycans and glycoproteins, which convey its unique mechanical viscoelastic

**Funding:** This study was funded by Bin Yan, National Natural Science Foundation of China (Grant No.81571005), the Key Technology R&D Program of Jiangsu Province (BE2018723), and the Priority Academic Program Development of Jiangsu Higher Education Institutions (PAPD, 2018-87) and also funded by Bin Wu, National Natural Science Foundation of China (Grant No. 51305208), Jiangsu Provincial Key Laboratory of Oral Diseases Research Fund (JSKLOD-KF-1901). The funders had no role in study design, data collection and analysis, decision to publish, or preparation of the manuscript.

**Competing interests:** The authors have declared that no competing interests exist.

properties [2]. In fact, the PDL composite structure is the basis for the manifestation of a non-linear and time-dependent mechanical behavior in this tissue [3–5].

It has been noted that experimental testing is an ideal method to study the physiological parameters of biological material [6]. Previously, the viscoelastic behavior of the PDL has been examined with uniaxial tensile [7] and stress relaxation tests [8, 9] using quasi-static experimental setups. Subsequently, the elastic behavior of the PDL has been demonstrated to be influenced by the loading rate, tooth type, root level and age. However, the human PDL exhibits different behavior when confronting dynamic circumstances such as mastication or non-physiological tooth movements. Oskui et al. [10] has characterized the short- and long-term dynamic tensile responses of the PDL within a specific frequency range. However, our knowledge on the dynamic viscoelastic properties of the human PDL is limited to tensile responses, which demand more investigations.

Due to collagen fibers' resistance against tensile loads and ground substance tolerance against compressive loads, the PDL has displayed different mechanical responses under tension and compressive loadings [11]. We clarified the viscoelastic properties of the human PDL under dynamic tension in our former study, in which the viscoelasticity of PDL was believed to be related to frequency, tooth position, and root level [12]. However, its mechanical behavior under dynamic compression loading is still to be determined. Therefore, an accurate quantification of mechanical properties of the human PDL under dynamic compressive loadings is required for deeper understanding of its mechanical behavior.

Dynamic mechanical analysis (DMA) is regarded as a powerful method to investigate the mechanical properties of biological materials, and it has been extensively employed in characterization of the viscoelastic properties of different tissues [13–15]. The viscoelastic properties of other tissues such as cardiovascular tissue [16] and articular cartilage [17, 18] have been investigated under dynamic loading. Compared to other methods, DMA is accounted for tendering more beneficial results due to its flexibility in changing various testing parameters and substantial accuracy [15]. Through application of dynamic oscillatory loadings, the storage modulus (E'), loss modulus (E") and loss factor (tanδ) of different biomaterials can be obtained via DMA. The storage modulus refers to the capacity to store energy in each cycle, which reflects the stiffness, while the loss modulus refers to the capacity to dissipate energy in each cycle, which reflects the viscosity [19–21]. The difference between the peak strain and the peak resulting stress is called phase lag, which is expressed as an angle (δ) ($0 < δ < 90°$ in viscoelastic systems). Tanδ, indicative of the ratio between the loss and storage modulus, denotes the mathematical description of the dynamic experiment under strains within a sample's range of viscoelasticity [5]. Accordingly, DMA enabled us to characterize the dynamic viscoelastic behavior of the human PDL.

The aim of this study was to quantify the viscoelastic properties of the human PDL under dynamic compressive loading and to determine the effects of loading frequencies and tooth location on the mechanical response of the PDL.

## Materials and methods

### Sample preparation

Institutional Review Board approval was granted from Nanjing Medical University prior to conduction of this experimental study (NO. (2015)169). The fresh corpses used in this study were provided by the Department of Anatomy of Nanjing Medical University with a declared legal consent given by the donor or their next of kin attained by the same administration. The study did not involve the use of donated tissue/organs from any vulnerable populations. The authors did not have access to any identifying information for the human samples. All

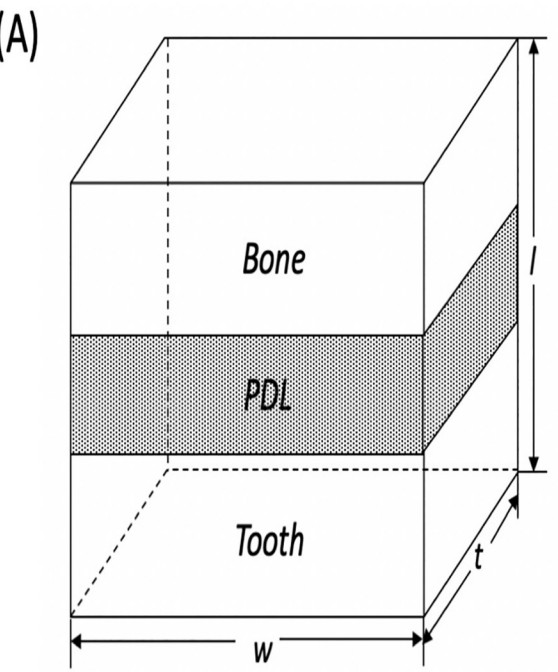

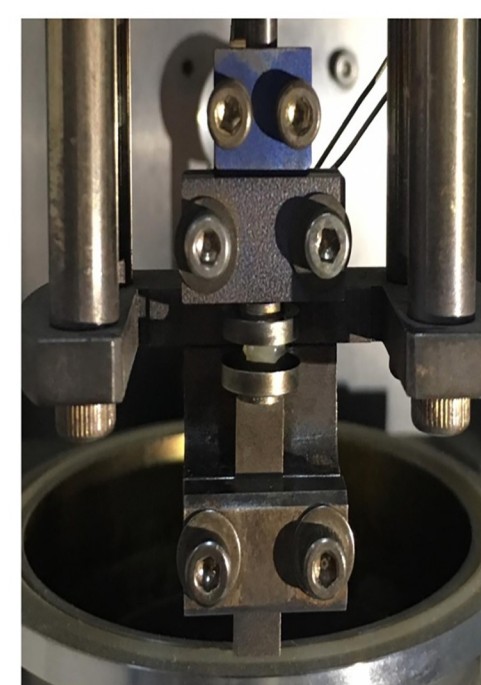

**Fig 1. (a) Schematic view of a cube-shaped sample; (b) compressive DMA setup for human PDL.**

specimens were fabricated using three human maxillary jaw segments of fresh corpses (male, 33–60 years old, dentally and periodontally healthy), consisting of 5 central and 5 lateral incisors. The authors had no knowledge of the sample's identity. The maxillary jaws were collected between 2017 to 2018, placed in a plastic container, and finally stored in a freezing container until the slicing procedure. The adherent soft tissues of the maxilla were manually peeled off from the jaw segments by a surgical scalpel. Each segment with incisor and alveolar bone was split up with a bone saw. Every unit composed of bone, PDL and cementum was supported by a fixture and embedded in wax for cutting into transverse sections. Transverse sections of 2-mm thickness were cut perpendicularly to the root longitudinal axis using a rotating blade saw (Isomet Low Speed Saw, Buehler, Lake Bluff, IL, USA) with 500 r/min velocity. Normal saline was profusely sprayed onto specimens while sawing to prevent heat-induced tissue denaturation. The experimental sample for each tooth was taken only from the transverse section of the cervical portion. The 2×2×2 sized cube-shaped samples (Fig 1a) were clipped from these sections using a dental low-speed handpiece. A total of 10 specimens were fabricated and preserved at -20˚C in normal saline solution until the biomechanical testing.

## Test machine

Dynamic mechanical analysis (DMA) is consigned for describing the physical properties of samples exposed to slight sinusoidal oscillating forces in relation to the frequency of the applied force, the temperature of the testing environment, and the exposure time. In the present study, DMA was implemented on human maxillary incisor PDL samples one week after the sample preparation. The DMA was applied to our prepared cube-shaped samples using a Pyris Diamond Dynamic Mechanical Analyzer (Perkin Elmer, Waltham, Massachusetts, U.S.) with a force resolution set at 0.2 mN and a displacement resolution set at 10 nm. Samples were

placed on the custom-designed plateaus (Fig 1b). The plateaus were coated with a thin layer of mineral oil to minimize the friction between the contact surfaces during the test procedure. The shear modulus of mineral oil was much lower than that of the tested bone–PDL–cementum complex so that the viscous influence of the mineral oil could be ignored.

## Test procedure

The test preconditions were achieved by thawing the samples at room temperature (21–24˚C) and submerging them in normal saline solution. All tests were carried out at room temperature.

To evaluate the viscoelastic properties of PDL, dynamic compression tests were performed. The induced sinusoidal stress from plateaus generated a sinusoidal strain on the testing sample, which actuated an angular difference between the peak load and sample deformation, expressed as phase lag (tanδ). Once the specimen was installed and the contact was established, a preload of 0.2 mN was applied to hold the specimen. Our pilot experiment confirmed a drastic increase in the dynamic moduli within 2 Hz frequency, and no apparent change in 5–10 Hz. Therefore, the loading frequency in the compression test ranged from 0.05 Hz to 5 Hz in the current experiment. Frequency sweep tests were performed between 0.05 Hz and 5 Hz (0.05, 0.1, 0.2, 0.5, 1, 2, 5 Hz) with a compression amplitude that was 2% of the PDL's initial width. The maximum strain of the PDL was set to be 8% of the PDL's initial width, which was adopted from a preliminary experiment. Minimal strain was applied to protect the PDL from fracture failure. Each specimen was tested in triplicate for 15 minutes. During test procedures, the experimental data became repeatable after 5 cycles [13].

## Data analysis

The dynamic compression parameters were determined by Fourier analyses for each frequency. By using Fourier transforms, the dynamic stiffness ($k^*$) was calculated from the magnitudes of the load ($F^*$) and displacement ($d^*$) according to Eq (1) [21]. Accordingly, storage modulus (E') and loss modulus (E") were then calculated based on Eqs (2) and (3), respectively [22]. The phase angle (δ) between the load and displacement signals was determined in Eq (4) [23].

$$k^* = \frac{F^*}{d^*} \tag{1}$$

$$E' = \frac{k^* \cos \delta}{S} \tag{2}$$

$$E'' = \frac{k^* \sin \delta}{S} \tag{3}$$

$$\tan \delta = \frac{E''}{E'} \tag{4}$$

S is the shape factor, which was calculated from Eq (5), whereas w is the width, t is the thickness, and l is the specimen length [21].

$$S = \frac{wt}{l} \tag{5}$$

To precondition the material and obtain repeatable data, several compression cycles were

**Table 1.  Mean and standard deviation of parameters measured over the frequency range of 0.05 Hz -5 Hz.**

| Frequency (*f*) | $E'$ (MPa) | | $E''$ (MPa) | | tan $\delta$ | |
|---|---|---|---|---|---|---|
| | Mean | SD | Mean | SD | Mean | SD |
| 0.05 | 96.9990 | 48.4854 | 28.3837 | 16.4158 | 0.2830 | 0.0558 |
| 0.1 | 112.5365 | 60.0797 | 32.8745 | 20.9342 | 0.2810 | 0.0597 |
| 0.2 | 133.6410 | 78.3974 | 36.5937 | 24.7863 | 0.2645 | 0.0568 |
| 0.5 | 162.5622 | 103.0892 | 38.7678 | 26.1856 | 0.2341 | 0.0481 |
| 1 | 178.4921 | 114.9279 | 38.8150 | 25.9102 | 0.2148 | 0.0442 |
| 2 | 194.0808 | 126.3554 | 38.7361 | 25.3360 | 0.1988 | 0.0382 |
| 5 | 220.4909 | 150.9015 | 39.2683 | 25.7419 | 0.1800 | 0.0318 |

conducted before data collection. After 5 loading cycles in the DMA test, experimental data from the last five cycles for each frequency were used to determine the biomechanical properties of the PDL. Considering the extremely high stiffness of the tooth and bone, we assumed the obtained data from the apparatus represent the dynamic compression properties of PDL.

All analyses were performed using IBM SPSS Statistics software (version 24, IBM, USA). Multifactor analysis of variance was utilized to compare the parameters of interest (storage modulus $E'$, loss modulus $E''$, loss tangent tan $\delta$) between different tooth locations and loading frequencies. The 95% confidence intervals were also generated using SPSS. The significance levels of the curve fit generated for storage and loss modulus were tested using regression analysis. All $p<0.05$ were considered statistically significant.

## Results

Generally, the storage modulus ranged from 28.61 MPa to 250.21 MPa. The value of the loss modulus was between 6.00 MPa and 49.28 MPa (Table 1). A significant difference was found in the storage modulus between different frequencies ($p<0.05$). The experimental results revealed that the storage modulus increased significantly as the frequency increased from 0.05 Hz to 0.5 Hz. The same trend was found for the loss modulus. However, the loss modulus between 1 Hz and 5 Hz remained almost constant, while the storage modulus still increased gradually with frequency. The storage modulus varied more observably than the loss modulus between the frequency range of 0.05 Hz-0.5 Hz (Fig 2). Conclusively, the loss factor (tan $\delta$) within the range of 0.1321 to 0.3587 demonstrated a significant negative relationship with the frequency ($p<0.05$), especially between 0.05 Hz to 1 Hz and the phase angle shift from 7.53˚ to 19.73˚.

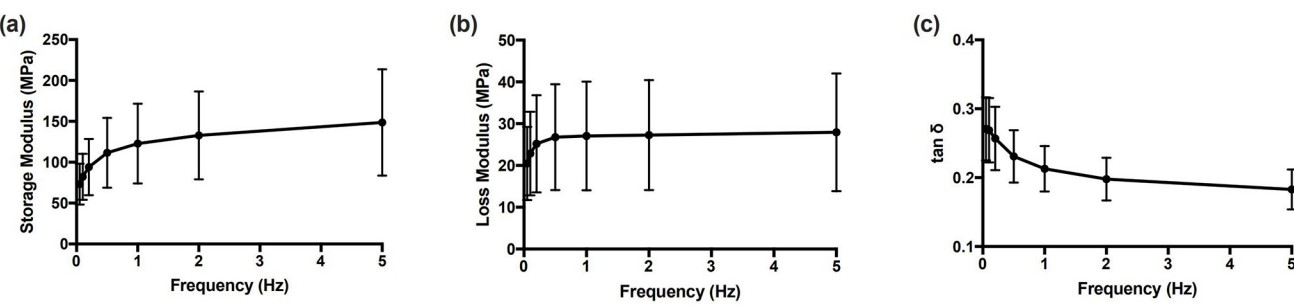

**Fig 2. Storage (E') modulus (a), loss (E") modulus (b) and tan$\delta$ (c) against frequency for all samples.** Error bars denote standard deviations.

**Table 2. Curve fits for the storage and loss tangents.** All coefficients were found to be statistically significant.

| Specimen Number | Storage modulus ($E'$) curve fit | | | Loss tangent (tan $\delta$) curve fit | | |
|---|---|---|---|---|---|---|
| | $E' = ae^{bf}$ | | | $\tan \delta = ce^{df}$ | | |
| | a | b | $R^2$ | c | d | $R^2$ |
| 1 (Lateral incisor) | 85.25 | 0.084 | 0.909 | 0.156 | -0.090 | 0.973 |
| 2 (Lateral incisor) | 44.55 | 0.125 | 0.782 | 0.183 | -0.073 | 0.870 |
| 3 (central incisor) | 115.65 | 0.104 | 0.940 | 0.183 | -0.119 | 0.938 |
| 4 (Lateral incisor) | 54.78 | 0.099 | 0.829 | 0.193 | -0.049 | 0.955 |
| 5 (Lateral incisor) | 103.41 | 0.179 | 0.925 | 0.246 | -0.053 | 0.794 |
| 6 (central incisor) | 179.50 | 0.184 | 0.956 | 0.246 | -0.085 | 0.840 |
| 7 (central incisor) | 162.78 | 0.162 | 0.917 | 0.205 | -0.108 | 0.873 |
| 8 (central incisor) | 180.08 | 0.185 | 0.954 | 0.245 | -0.086 | 0.854 |
| 9 (central incisor) | 132.82 | 0.161 | 0.647 | 0.234 | -0.133 | 0.894 |
| 10 (Lateral incisor) | 132.85 | 0.186 | 0.930 | 0.243 | -0.125 | 0.923 |

An exponential function (Eq (6)) could describe the storage modulus variation over the frequency range of 0.05 Hz to 5 Hz:

$$E' = 108.661e^{0.147f} \text{ for } 0.05 < f < 5 \tag{6}$$

in which $f$ stands for frequency.

Based on regression analysis, the curve fits with $R^2$ values of $\geq 0.65$ (Table 2) explained a significant strong correlation (Fig 3a). Negative correlation between frequency and the loss tangent was found, which could be portrayed by an exponential function as:

$$\tan \delta = 0.211e^{0.092f} \text{ for } 0.05 < f < 5 \tag{7}$$

in which $f$ stands for frequency.

The curve fit showed strong correlation with $R^2$ values of $\geq 0.794$ (Table 2), which likewise were statistically significant (Fig 3a). Contrarily, similar trends were not found in the loss modulus since there was no significant difference when different testing frequencies were used ($p > 0.05$). The loss modulus remained almost constant from 0.5 Hz-5 Hz in each specimen.

The experimental results of the central incisor group were significantly different ($p < 0.05$) from those of the lateral incisor group in storage and loss moduli (Fig 4). The mean value of the storage modulus in the central incisor group was 141.21 MPa, while the corresponding

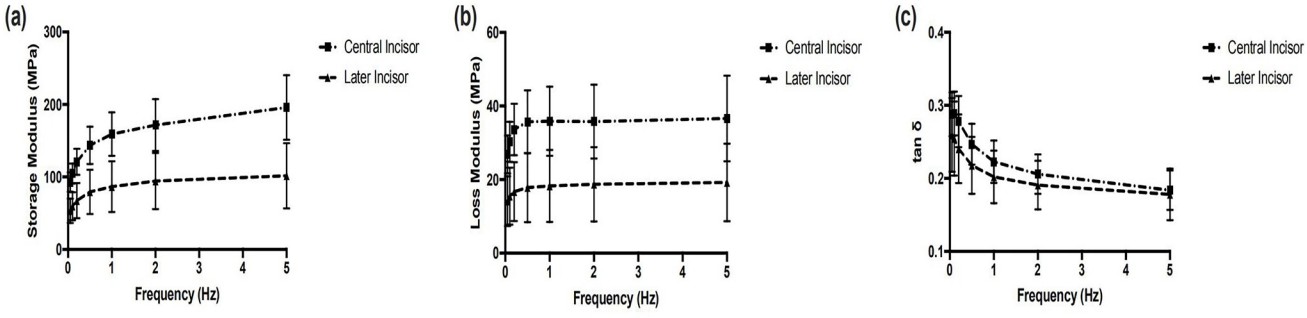

**Fig 3. Fitting curves of storage modulus (a) and tan$\delta$(b) for all human PDL specimens.** Data points represent the respective mean values for the specimens.

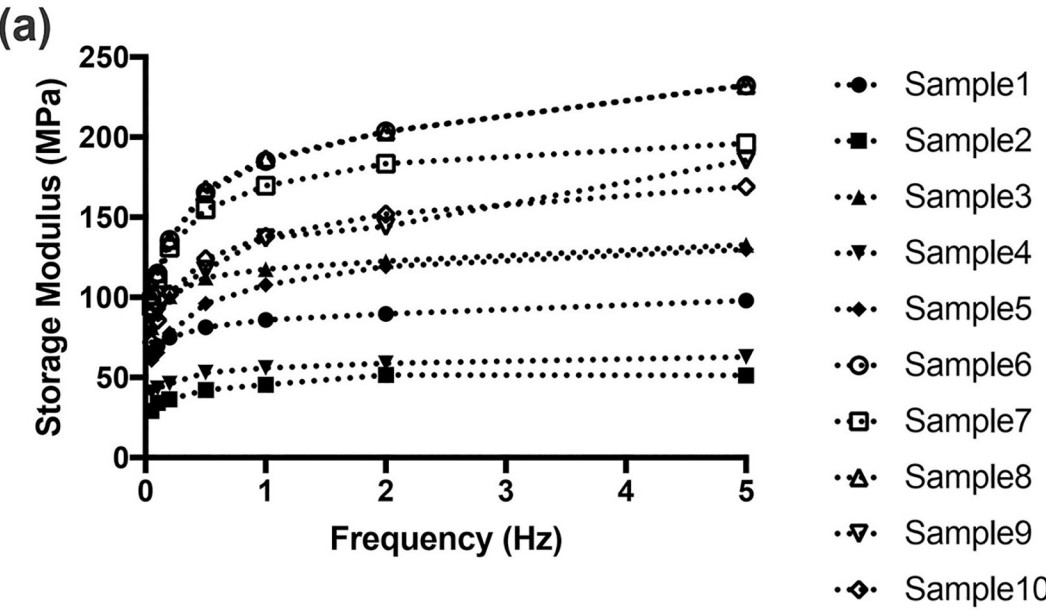

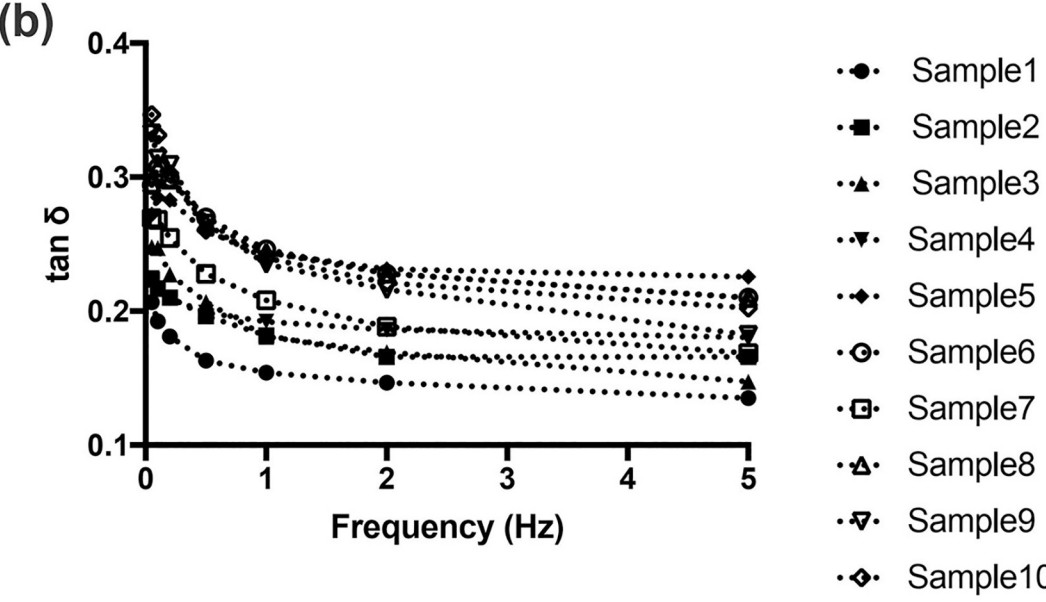

**Fig 4. The storage modulus, loss modulus and tan$\delta$ of the central and lateral incisors.**

value of the lateral incisor group was 77.49 MPa. The mean values of the loss modulus of the central and lateral incisor groups were 33.53 MPa and 17.20 MPa, respectively (Table 3). Comparison between the two groups also revealed that the loss tangent of the central incisor group was higher than that of the lateral incisor group.

**Table 3. Storage and loss moduli and tan$\delta$ for central incisors and lateral incisors.**

| Frequency ($f$) | Central incisor | | | | | | Lateral incisor | | | | | |
|---|---|---|---|---|---|---|---|---|---|---|---|---|
| | $E'$ (MPa) | | $E''$ (MPa) | | tan $\delta$ | | $E'$ (MPa) | | $E''$ (MPa) | | tan $\delta$ | |
| | Mean | SD | Mean | SD | Mean | SD | Mean | SD | Mean | SD | Mean | SD |
| 0.05 | 92.99 | 13.43 | 26.83 | 5.14 | 0.288 | 0.030 | 53.39 | 16.87 | 14.11 | 6.77 | 0.255 | 0.0531 |
| 0.1 | 104.59 | 13.81 | 30.25 | 5.45 | 0.289 | 0.030 | 59.67 | 19.25 | 15.46 | 7.75 | 0.248 | 0.0532 |
| 0.2 | 120.81 | 18.18 | 33.65 | 7.04 | 0.278 | 0.035 | 67.45 | 24.37 | 16.74 | 8.01 | 0.237 | 0.0484 |
| 0.5 | 143.60 | 25.89 | 35.73 | 8.57 | 0.247 | 0.028 | 79.36 | 30.48 | 17.84 | 9.39 | 0.216 | 0.0398 |
| 1 | 159.16 | 30.03 | 35.88 | 9.42 | 0.223 | 0.029 | 86.63 | 34.98 | 18.29 | 9.80 | 0.202 | 0.0359 |
| 2 | 171.51 | 35.83 | 35.75 | 10.08 | 0.206 | 0.027 | 94.30 | 38.94 | 18.74 | 10.13 | 0.190 | 0.0336 |
| 5 | 195.84 | 44.63 | 36.62 | 11.66 | 0.184 | 0.027 | 101.64 | 45.00 | 19.23 | 10.58 | 0.182 | 0.0319 |

## Discussion

The human PDL is known as a heterogenous connective tissue mainly composed of fiber networks and bundles with different orientations and locations longitudinally. The PDL fibers around the cervical part of the tooth root are oriented horizontally, while the PDL in the middle part of the root consists of the oblique fibers [24]. An earlier study reported that as a result of sawing, the disruption of oblique fibers in the PDL within the transverse section can cause inevitable errors [25]. Therefore, to avoid damage of shearing forces under axial compression and to minimize the side effects of specimen preparation, the cervical PDL was chosen for the current study. It has been reported that mechanical properties of parallel-fibered specimens, such as ligaments, arteries, and tendons, stored at -10˚C to -20˚C changed negligibly [6]. Hence, in this study, the irreversible alterations resulting from freezing as a storage protocol were considered insignificant and ignorable.

The storage and loss moduli are respectively designating the elasticity and viscosity properties of the PDL. As shown in Fig 2, the storage modulus increased significantly in the range of 0.05 Hz to 0.5 Hz and increased slightly from 1.0 Hz to 5 Hz. The loss modulus also ascended with the frequency increase from 0.05 Hz to 0.5 Hz. However, it remained almost constant from 1.0 Hz to 5 Hz. The trends found in storage and loss moduli revealed that the viscoelasticity of periodontal ligament is correlated with the frequency. Consistent with previous findings [4], increasing the frequency caused a stiffer and less compressible PDL with more viscosity (0.05 Hz-0.5 Hz), which may hamper the desired physical distortion and deformation of the PDL when an orthodontic force is applied. The loss factor (tan $\delta$) (0.3587 to 0.1321) had a significant negative relationship with the frequency, especially between 0.05 Hz and 1 Hz during dynamic compression, suggesting that PDL demonstrates elasticity mainly in high-frequency loadings, i.e., instantaneous response.

Consistent with previous findings of the human PDL revealing a frequency-related viscoelasticity under dynamic tension [12], our current experiment yielded similar results when the analysis was extended to compressive forces. However, the dynamic moduli were much greater in the compression test than in the tension test (E' from 0.808 MPa to 7.274 MPa, E" from 0.087 MPa to 0.891 MPa). Additionally, the phase lag and hysteresis in compression were much higher. Primarily, these differences may be the result of different loading conditions between the tension and compression tests, such as the size of the PDL (tension test is 8×6×2 mm, compression test is 2×2×2 mm). [26] Li et al [27] perceived that the viscoelasticity of the periodontal ligament is related to the direction of loading. In the tension test, the collagen fibers played a dominant role in the mechanical response. In the compression test, the behavior of the PDL seemed to be influenced by interstitial fluid flow and can be considered as a

biphasic material, such as articular cartilage [27–29]. Considering the different major contributing elements during tension and compression, the mechanical prosperity of the PDL during strain or condensation can be anticipated, which requires further study. These findings indicate that the PDL might exhibit different biomechanical reactions under tension and compression, affecting the tooth intrusion and extrusion movements.

The experimental results showed that the values of the storage and loss moduli in the central incisor group were almost triple of those in the lateral incisor group, meaning that the PDL of the central incisor was stiffer, with a higher viscosity than that of the lateral incisor. Thus, the central incisor needs more force than the lateral incisor to accomplish tooth movement, which is consistent with clinical observations. It can be proposed that in addition to the PDL proportion when choosing anchorage teeth, the stiffness and viscosity of the PDL should also be considered. In concordance with Tanaka [30], the higher viscosity of the central incisor in our study demonstrated that the central incisor PDL has a better energy-absorbing or shock-absorbing capacity for sudden loading than the lateral incisor, indicating that the central incisor PDL can withstand more masticatory force than the lateral incisor PDL. However, the central and the lateral incisors are adjacent teeth and are often under similar mastication load and retracted as a whole during orthodontic treatment.

Our findings may provide instrumental information for orthodontic treatment and tooth movement simulation. According to the present results, when exaggerated orthodontic loading frequency was applied, the PDL became stiffer, which is an undesirable outcome for orthodontic tooth movement. Hence, application of light orthodontic force is crucial in orthodontic treatment. The dynamic parameters, E', E" and tanδ, changed markedly in the frequencies between 0.05 Hz and 0.5 Hz. This finding proves that the dynamic loading under specific frequency evidently changes the viscoelasticity of the PDL, which is helpful for controlling or accelerating the orthodontic treatment. In addition, the compression experimental data are important for numerical simulations and will be complementary for establishing constitutive models of periodontal ligament, which can be used to simulate the tooth movement in orthodontic treatment.

There are possible limitations that should be considered. The present study has only measured dynamic parameters under a low level of axial compression stress. The masticatory force is much higher than the experimental stress. The current experiments attempted to determine the compression properties of whole intact PDL. However, the results only reflected the mechanical response of the bone-PDL-tooth complex. The data typically exhibited a large standard deviation, which necessitates multiple specimens in future research.

## Conclusion

This study illustrated that the human PDL exhibits viscoelastic behavior under compressive loadings within the range of the used frequency, 0.05 Hz– 5 Hz. The tooth position and testing frequency may have effects on the viscoelastic properties of the PDL. The results of this study will benefit researchers and orthodontists by providing a deeper and more profound understanding of the mechanical properties and will provide complementary information for FE simulation of orthodontic treatment.

## Supporting information

**S1 Data.**
(ZIP)

## Author Contributions

**Conceptualization:** Bin Wu, Siyu Zhao, Bin Yan, Songyun Ma, Bernd Markert.

**Data curation:** Panjun Pu, Siyu Zhao, Ruxin Lu.

**Formal analysis:** Bin Wu, Panjun Pu, Siyu Zhao, Iman Izadikhah, Ruxin Lu.

**Funding acquisition:** Bin Wu, Bin Yan.

**Methodology:** Bin Wu, Panjun Pu, Siyu Zhao, Iman Izadikhah, Haotian Shi, Bernd Markert.

**Project administration:** Bin Wu, Panjun Pu, Bin Yan.

**Resources:** Ruxin Lu, Bin Yan.

**Software:** Panjun Pu, Haotian Shi, Mao Liu, Ruxin Lu.

**Supervision:** Mao Liu, Bin Yan, Songyun Ma.

**Validation:** Iman Izadikhah, Haotian Shi, Bin Yan, Songyun Ma, Bernd Markert.

**Visualization:** Bin Wu, Haotian Shi, Mao Liu, Bin Yan, Bernd Markert.

**Writing – original draft:** Panjun Pu, Siyu Zhao, Iman Izadikhah.

**Writing – review & editing:** Iman Izadikhah.

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
