## [Decision Letter · Decision Letter 0]

9 Apr 2020

PONE-D-19-34657

Frequency-related viscoelastic properties of human incisor periodontal ligament under dynamic compressive loading

PLOS ONE

Dear Dr. Yan,

Thank you for submitting your manuscript to PLOS ONE. After careful consideration, we feel that it has merit but does not fully meet PLOS ONE’s publication criteria as it currently stands. Therefore, we invite you to submit a revised version of the manuscript that addresses the points raised during the review process.

We would appreciate receiving your revised manuscript by May 24 2020 11:59PM. To enhance the reproducibility of your results, we recommend that if applicable you deposit your laboratory protocols in protocols.io, where a protocol can be assigned its own identifier (DOI) such that it can be cited independently in the future. For instructions see: http://journals.plos.org/plosone/s/submission-guidelines#loc-laboratory-protocols

We look forward to receiving your revised manuscript.

Kind regards,

Jose Manuel Garcia Aznar

Academic Editor

PLOS ONE

Journal Requirements:

1. In your Methods section, please provide additional information regarding the human samples used in the study. Specifically:

- Please specify the source of the fresh corpses.

- State in your ethics statement whether the study involved the use of donated tissue/organs from any vulnerable populations.

- Provide information on the consent given by the donor or their next of kin. Examples of vulnerable populations include prisoners, subjects with reduced mental capacity due to illness or age, and children. If a vulnerable population was used, please describe the population and justify the decision to use tissue/organ donations from this group. If not, please state in your Ethics Statement, "None of the corpses were from a vulnerable population and all donors or next of kin provided written informed consent that was freely given."

- If the authors did not have access to any identifying information for the human samples, please state this.

Reviewers' comments:

Reviewer's Responses to Questions

**Comments to the Author**

1. Is the manuscript technically sound, and do the data support the conclusions?

Reviewer #1: Yes

2. Has the statistical analysis been performed appropriately and rigorously? 

Reviewer #1: Yes

3. Have the authors made all data underlying the findings in their manuscript fully available?

Reviewer #1: Yes

4. Is the manuscript presented in an intelligible fashion and written in standard English?

Reviewer #1: Yes

5. Review Comments to the Author

Reviewer #1: In the paper, the results of dynamic compression tests are presented for the periodontal human ligament. The study aims to determine the viscoelastic properties of this tissue which have not been previously analyzed in the literature. The manuscript is well written, structured and the discussion section provides interesting theories and perspectives of the ligament biomechanical role.

As a general comment regarding the way the results are presented, why Table 1 and Fig. 2 include the statistics of all the samples and then they are split into central and lateral incisors? Why not presenting then separately from the beginning? I couldn’t find an explanation of this fact in the text.

6. PLOS authors have the option to publish the peer review history of their article (what does this mean?). If published, this will include your full peer review and any attached files.

Reviewer #1: No

---

## [Author Response · Author response to Decision Letter 0]

22 May 2020

Response to Reviewers

We appreciate the reviewers’ positive comments and constructive suggestions on our manuscript, entitled “Frequency-related viscoelastic properties of the human incisor periodontal ligament under dynamic compressive loading” (PONE-D-19-34657). We have addressed all of the comments to the best of our ability, and we believe the revisions requested by the reviewers have significantly improved the quality of our manuscript. The reviewers’ unedited comments are provided below in black text followed by our responses in blue text. 

1. Authors’ reply: 

Thank you for your reminder. We have added another fund named Jiangsu Provincial Key Laboratory of Oral Diseases Research Fund (JSKLOD-KF-1901) which has also been stated it in our cover letter. 

2. To enhance the reproducibility of your results, we recommend that if applicable you deposit your laboratory protocols in protocols.io, where a protocol can be assigned its own identifier (DOI) such that it can be cited independently in the future. For instructions see: http://journals.plos.org/plosone/s/submission-guidelines#loc-laboratory-protocols.

3. In your Methods section, please provide additional information regarding the human samples used in the study. Specifically:

- Please specify the source of the fresh corpses.

- State in your ethics statement whether the study involved the use of donated tissue/organs from any vulnerable populations.

- Provide information on the consent given by the donor or their next of kin. Examples of vulnerable populations include prisoners, subjects with reduced mental capacity due to illness or age, and children. If a vulnerable population was used, please describe the population and justify the decision to use tissue/organ donations from this group. If not, please state in your Ethics Statement, "None of the corpses were from a vulnerable population and all donors or next of kin provided written informed consent that was freely given."

- If the authors did not have access to any identifying information for the human samples, please state this.

Authors’ reply: 

Thank you for your valuable review of our work. The fresh corpses came from Department of Anatomy of Nanjing Medical University with a declared legal consent given by the donor or their next of kin attained by the same administration. The study did not involve the use of donated tissue/organs from any vulnerable populations. We did not have access to any identifying information for the human samples. 

More details have been provided in the revised paper, as listed below: 

The fresh corpses used in this study were provided by the Department of Anatomy of Nanjing Medical University with a declared legal consent given by the donor or their next of kin attained by the same administration. The study did not involve the use of donated tissue/organs from any vulnerable populations. The authors did not have access to any identifying information for the human samples. (Details see page 6 line 119-123.)

Authors’ reply: 

Thank you for your valuable suggestion. We apologize the language issues in the previous manuscript. To make our paper easier to follow, the manuscript has been extensively reviewed and revised by American Journal Experts (AJE). We hope that the language in the revised paper has been substantially improved.

Reviewer #1 As a general comment regarding the way the results are presented, why Table 1 and Fig. 2 include the statistics of all the samples and then they are split into central and lateral incisors? Why not presenting then separately from the beginning? I couldn’t find an explanation of this fact in the text.

Authors’ reply: 

Thank you for your valuable suggestion. Because the central incisor and the lateral incisor are adjacent teeth, and they are under similar bite force. Moreover, the central incisor and the lateral incisor are often retracted as a whole in orthodontic treatment. Therefore, we showed the results including the central incisor and the lateral incisor at the beginning to show the effect of frequency on the viscoelastic properties of the incisor PDL. 

However, in a more detailed and specified view, the higher viscosity of the central incisor demonstrated that its PDL has a better energy-absorbing or shock-absorbing capacity for sudden loading than the lateral incisor, which indicates that the central incisor PDL can withstand more mastication force than the lateral incisor PDL. Since this particular comparison between the incisors in our study was of importance during orthodontic force application, we presented the central incisor and lateral incisor separately additionally to the scope of our study.

More details have been provided in the revised paper, as listed below: 

In concordance with Tanaka [30], the higher viscosity of the central incisor in our study demonstrated that the central incisor PDL has a better energy-absorbing or shock-absorbing capacity for sudden loading than the lateral incisor, indicating that the central incisor PDL can withstand more bite force than the lateral incisor PDL. However, the central incisor and the lateral incisor are adjacent teeth and are often under similar bite force and retracted as a whole during orthodontic treatment (for details, see page 16, lines 311-317.)

---

## [Decision Letter · Decision Letter 1]

24 Jun 2020

Frequency-related viscoelastic properties of human incisor periodontal ligament under dynamic compressive loading

PONE-D-19-34657R1

Dear Dr. Yan,

We’re pleased to inform you that your manuscript has been judged scientifically suitable for publication and will be formally accepted for publication once it meets all outstanding technical requirements.

Kind regards,

Jose Manuel Garcia Aznar

Academic Editor

PLOS ONE

Additional Editor Comments (optional):

Reviewers' comments:

Reviewer's Responses to Questions

**Comments to the Author**

1. If the authors have adequately addressed your comments raised in a previous round of review and you feel that this manuscript is now acceptable for publication, you may indicate that here to bypass the “Comments to the Author” section, enter your conflict of interest statement in the “Confidential to Editor” section, and submit your "Accept" recommendation.

Reviewer #1: All comments have been addressed

2. Is the manuscript technically sound, and do the data support the conclusions?

Reviewer #1: Yes

3. Has the statistical analysis been performed appropriately and rigorously? 

Reviewer #1: Yes

4. Have the authors made all data underlying the findings in their manuscript fully available?

Reviewer #1: No

5. Is the manuscript presented in an intelligible fashion and written in standard English?

Reviewer #1: Yes

6. Review Comments to the Author

Reviewer #1: (No Response)

7. PLOS authors have the option to publish the peer review history of their article (what does this mean?). If published, this will include your full peer review and any attached files.

Reviewer #1: No

---

## [Editor Report · Acceptance letter]

1 Jul 2020

PONE-D-19-34657R1 

Frequency-related viscoelastic properties of the human incisor periodontal ligament under dynamic compressive loading 

Dear Dr. Yan:

I'm pleased to inform you that your manuscript has been deemed suitable for publication in PLOS ONE. Congratulations! Your manuscript is now with our production department. 

Kind regards, 

on behalf of

Dr. Jose Manuel Garcia Aznar 

Academic Editor

PLOS ONE